# CaHSP18.1a, a Small Heat Shock Protein from Pepper (*Capsicum annuum* L.), Positively Responds to Heat, Drought, and Salt Tolerance

Yan-Li Liu [1,†], Shuai Liu [1,†], Jing-Jing Xiao [1], Guo-Xin Cheng [1], Haq Saeed ul [1,2] and Zhen-Hui Gong [1,*]

[1] College of Horticulture, Northwest A&F University, Yangling 712100, China; 2018060114@nwafu.edu.cn (Y.-L.L.); liushuai5887@nwsuaf.edu.cn (S.L.); 2018050339@nwafu.edu.cn (J.-J.X.); 2014060108@nwsuaf.edu.cn (G.-X.C.); Saeed_ulhaq@nwsuaf.edu.cn (H.S.u.)

[2] Department of Horticulture, University of Agriculture Peshawar, Peshawar 25120, Pakistan

[*] Correspondence: zhgong@nwsuaf.edu.cn; Tel.: +86-029-8708-2102; Fax: +86-029-8708-2613

[†] Co-first author.

**Abstract:** Pepper is a thermophilic crop, shallow-rooted plant that is often severely affected by abiotic stresses such as heat, salt, and drought. The growth and development of pepper is seriously affected by adverse stresses, resulting in decreases in the yield and quality of pepper crops. Small heat shock proteins (s HSPs) play a crucial role in protecting plant cells against various stresses. A previous study in our laboratory showed that the expression level of *CaHSP18.1a* was highly induced by heat stress, but the function and mechanism of CaHSP18.1a responding to abiotic stresses is not clear. In this study, we first analyzed the expression of *CaHSP18.1a* in the thermo-sensitive B6 line and thermo-tolerant R9 line and demonstrated that the transcription of *CaHSP18.1a* was strongly induced by heat stress, salt, and drought stress in both R9 and B6, and that the response is more intense and earlier in the R9 line. In the R9 line, the silencing of *CaHSP18.1a* decreased resistance to heat, drought, and salt stresses. The silencing of *CaHSP18.1a* resulted in significant increases in relative electrolyte leakage (REL) and malonaldehyde (MDA) contents, while total chlorophyll content decreased under heat, salt, and drought stresses. Overexpression analyses of *CaHSP18.1a* in transgenic *Arabidopsis* further confirmed that *CaHSP18.1a* functions positively in resistance to heat, drought, and salt stresses. The transgenic *Arabidopsis* had higherchlorophyll content and activities of superoxide dismutase, catalase, and ascorbate peroxidase than the wild type (WT). However, the relative conductivity and MDA content were decreased in transgenic Arabidopsis compared to the wild type (WT). We further showed that the CaHSP18.1a protein is localized to the cell membrane. These results indicate CaHSP18.1a may act as a positive regulator of responses to abiotic stresses.

**Keywords:** *CaHSP18.1a*; gene silencing; transgenic *Arabidopsis*; heat stress; pepper; gene expression

## 1. Introduction

Plants can tolerate considerable biotic and abiotic stresses in their complex and changing environments, including drought, high salt, extreme temperatures, and oxidation [1,2]. To mitigate stresses, plants have developed several protective mechanisms. Heat shock proteins (HSPs) can maintain protein homeostasis and prevent or repair the misfolding of proteins in abiotic stresses response. Moreover, HSPs are evolutionarily conserved molecular chaperones widely found among various plant taxa [3–5]. Plant HSPs also play critical roles in the folding, transport, degradation, and assembly of proteins under normal and stress conditions [6]. In response to high temperatures, plant cells dramatically increase the concentrations of HSPs to prevent heat-related damage and increase plant thermotolerance [7]. In addition, HSPs are also involved in plant growth and development under normal conditions, including the growth of flowers and seeds as well as fruit set, development [8], tuberization [9], and nutrient uptake [10]. HSPs are present in the cell membrane

and cytoplasm, nucleus, and cell organelles such as the mitochondria, chloroplasts, and endoplasmic reticulum [11,12].

HSPs, based on their sequence homology and molecular weight, are generally grouped into the following different families: HSP20s, HSP60s, HSP70s, HSP90s, and HSP100s [13,14]. Of the five conserved families, HSP20s, are also called small heat shock proteins (s HSPs). The molecular weights of HSP20s range between 15 and 42 kDa [13,15,16]. Furthermore, one of the distinctive characteristics of HSP20s is their ability to bind to substrate proteins without ATP, and they also have a strong ability to bind to denatured substrates [15–18]. Thus, s HSPs are highly able to maintain the stability of foreign proteins in cells to prevent them from aggregating. Although there are many types of substrate proteins, s HSPs have a flexible N-terminus and $\alpha$-crystallin domain (ACD) hydrophobic surface that can adapt to bind these different protein substrates. In addition, s HSPs can be combined with different substrates in different ways, which makes s HSPs able to bind to a wider variety of proteins and to provide more complicated mechanisms of action among HSPs [19].

Korotaeva et al. [20] and Nieto-Sotelo et al. [21] showed that different HSPs are differentially expressed in different species, and even among different genotypes of the same species. It has been reported that the overexpression of AtHSP17. 6A increased the penetration resistance of *Arabidopsis* [22]. *AtHSP21* improved the heat resistance of transgenic *Arabidopsis* and extended the memory time of plants subjected to heat resistance, such that *Arabidopsis* was more heat resistant when subjected to heat stress again [23]. Some studies have also reported that HSP gene expression positively regulated protective enzyme activities. For example, in *Arabidopsis*, overexpression of *AtHSP17.8* enhanced SOD activity [24]. Similarly, overexpression of *HSP16.9* in tobacco increased the activities of POD, CAT, and SOD [25].

Pepper (*Capsicum annuum* L.) is one of the most important economical and medicinal vegetable crops worldwide [26].Pepper is usually cultivated in warm regions under temperatures of 15–34 °C [27]. Salt, drought, and heat stress can limit pepper growth and development and severely damage pepper pollination and seed set, which can lead to flower and fruit abscission and thus lower pepper fruit yield and quality [28,29].HSP20s in pepper play a major role in environmental stress responses, and a total of 35 pepper HSP20s were identified by Guo et al. [30]. All HSP20s were named based on their molecular weights, and stress-related cis-elements were detected in the promoter regions, including heat shock elements (HSEs), TATA boxes, CCAAT motifs, and TC-rich repeats [26]. Many CaHSP20 genes are not expressed across different pepper tissues (i.e., root, stem, leaf, and flower tissues). In recent years, the functions of CaHSP22.4, CaHSP25.9, CaHSP16.4, CaHSP24.2, and CaHSP26 have been identified. CaHSP16.4 is localized to the cytoplasm and nucleus, while in *Arabidopsis* lines with *CaHSP16.4* overexpression, increased tolerance to heat stress has been observed [31]. Guo et al. [26] also found that overexpression of CaHSP22.4, which is located in the mitochondria and cytoplasm, increased heat tolerance in *Arabidopsis*, with the expression increasing when pepper plants were subjected to high temperature. Similarly, the CaHSP25.9 protein was localized to the cell membrane and cytoplasm, and positively regulates heat, salt, and drought stress tolerance in pepper (*Capsicum annuum* L.) [32]. Pepper CaHSP24.2 is localized to mitochondria, the cytoplasm, and chloroplasts, where *CaHSP24.2* enhances the thermo-tolerance of transgenic *Arabidopsis* plants and regulates the expression of heat stress-related genes [30]. He et al. [33] overexpressed *CaHSP26*, which enhanced the tolerance of heat stress in *Arabidopsis*. Interestingly, heat-tolerance and salt-tolerance decreased in *CaHSP22.0*-silenced pepper [34]. All these studies suggest that sHSP20s may participate in responses to heat stress [35] and contribute to the acquisition of pepper thermo-tolerance [30].

Among the 35 *CaHSP20s* examined, the expression level of *CaHSP18.1a* was increased in both the B6 and R9 lines under heat stress [30]. Moreover, sequence analysis showed that *CaHSP18.1a* contained an HSE, and some other stress-related elements were also identified [30]. Based on the above findings, we analyzed the subcellular localization and expression pattern of *CaHSP18.1a* in different pepper tissues, as well as its response to salt,

drought, and heat stresses. Virus-induced gene silencing (VIGS) was preliminarily used to analyze the functions of *CaHSP8.1a* in response to stress in pepper plants. In addition, overexpression (OE) in transgenic *Arabidopsis thaliana* indicated that *CaHSP18.1a* plays a positive regulatory role in the responses to heat, salt, and drought stress. Our results provide a basis for further functional studies of *CaHSP18.1a* in other important crop species and in its role in stress tolerance.

## 2. Materials and Methods

### 2.1. Plant Materials and Growth Conditions

The thermo-tolerant pepper line R9 (a sweet pepper from the World/Asia Vegetable Research and Development Center, PP0042-51) and the thermo-sensitive pepper line B6 (selected by the Pepper Research Group, College of Horticulture, Northwest A&F University, Yangling, China) were used in this study. Pepper seedlings were cultivated in a growth chamber under the following growth conditions prior to various treatments: daily 16 h light/8 h dark cycles and 65% relative humidity until the 6–8 true leaves stage. The temperature was changed throughout the course of the experiment. R9 peppers were grown under 25/20 °C day/night temperatures to enable analyzing gene expression [30,36,37]. However, the growing conditions for use of virus-induced gene silencing (VIGS) in the R9 pepper line were 22/18 °C day/night temperatures [38]. *Arabidopsis* ecotype Col-0 variety seedlings were incubated at 65% relative humidity, 22/18 °C (day/night), and 16 h/8 h (light/dark) photoperiod conditions [37,38].

### 2.2. RNA Extraction and Real-Time Fluorescent Quantitative PCR qRT-PCR Analysis

Total RNA was extracted using the Trizol method [28]. Synthesis of cDNA was conducted with the PrimeScript™ kit (Takara, Dalian, China) according to the manufacturer's instructions. First, we downloaded the amino acid sequence of *CaHSP18.1a* from the Pepper Genomics Database (accessed date on 1 January 2020, http://peppergenome.snu.ac.kr/: Accession number: CA08g17060). Primer Premier 5.0 was used to design primers, and primer specificity was detected using NCBI Primer BLAST (https://www.ncbi.nlm.nih.gov/tools/primer-blast/, accessed on 5 January 2020) (Supplementary Table S1). The pepper ubiquitin binding gene *CaUbi3* (Accession number AY486137) was used as a reference gene [39]. qRT-PCR was performed using the iQ5.0 Bio-Rad iCycler thermal cycler (Bio-Rad, Hercules, CA, USA). The SYBR Green Super mix (Takara, Dalian, China) was used in the qRT-PCR reaction system following the manufacturer's instructions. The relative expression levels of the gene were analyzed using the $2^{-\Delta\Delta CT}$ method [40].

### 2.3. Subcellular Localization of CaHSP18.1a Protein

The ORF (open reading frame) of *CaHSP18.1a* without a termination codon was PCR-amplified using a specific primer pair (Supplementary Table S1). The resulting *CaHSP18.1a* fragment was cloned into the pVBG2307: GFP vector with *Xba*I and *Kpn*I restriction sites. The pVBG2307:*CaHSP18.1a*: GFP fusion protein transient expression vector and the control vector pVBG2307: GFP, after having been successfully constructed, were transformed into *Agrobacterium tumefaciens* strain GV3101, which was then injected into tobacco (*Nicotiana tabacum*) leaves to induce transient expression. After dark cultivation for approximately 36 h, epidermis samples of tobacco leaves were photographed under a fully automatic upright fluorescence microscope on the public platform of the College of Horticulture, Northwest A&F University, and the fluorescence patterns in the cells were observed; we specifically used the method described by Yu et al. [41].

### 2.4. Virus-Induced Gene Silencing of CaHSP18.1a

A 256-bp fragment of the *CaHSP18.1a* ORF was PCR-amplified using a specific primer pair (Supplementary Table S1). The underlined sequences are restriction enzyme cleavage sites (for *Xba*I and *Kpn*I). The resulting *CaHSP18.1a* fragment was inserted into TRV2:00 vectors, with the empty vector TRV2:00 and TRV2: *CaPDS* (phytoene desaturase gene) used

as negative and positive controls. When R9 plants reached the two true leaves stage, we followed the method of Wang et al [38], which involved mixing the pTRV1 bacterial culture with an equal volume of the TRV2:00, TRV2: *CaPDS*, and TRV2: *CaHSP18.1a* cultures; this solution was injected into the leaves of R9 plants. After incubation in the dark at 18 °C for 2 days, plants were transferred to incubators under preset normal conditions. After 35 days, when most of the leaves of the TRV2-*CaPDS* pepper plants had become bleached, total RNA was extracted from the leaves of the silenced TRV2: *CaHSP18.1a* plants and the negative control TRV2:00 plants, and qRT-PCR was used to detect the *CaHSP18.1a* expression level, which was used to calculate silencing efficiency.

### 2.5. Generation of CaHSP18.1a-Overexpression Arabidopsis Lines

The entire coding regions of *CaHSP18.1a* were cloned into the pVBG2307 vector between the *Xba*I and *Kpn*I restriction sites to yield the final plasmid pVBG2307: *CaHSP18.1a* used for genetic transformation (the primers used for this experiment are given in Supplementary Table S1). The recombinant fusion vector was transformed into *Agrobacterium* strain GV3101 and transformed into *Arabidopsis thaliana* as described by Clough and Bent [42]. Transformed strains of pVBG2307 expression vector were screened with kanamycin and confirmed by PCR verification. We extracted DNA to detect the correctness of the target band. First, the fragment lengths of the bands were compared with the target band, obtaining, respectively, the OE1, OE2, OE3, OE4, and OE5 lines. Next, we performed real-time qRT-PCR quantitative analysis and detected the transcript from the inserted construct (Supplementary Figure S2B). *CaHSP18.1a* was thus determined to be expressed in large quantities in the OE3, OE2, and OE1 strains, but the wild-type gene was not detected. Both the target band and qRT-PCR results indicated that the *CaHSP18.1a* gene was successfully transferred into *Arabidopsis thaliana*, and the obtained T3-generation *Arabidopsis thaliana* could thus be used for further experiments.

### 2.6. Experimental Treatments and Sample Collection

The roots, stems, and leaves of R9 and B6 pepper seedlings (at the 4-to-6-leaf stage) grown under normal conditions were sampled in order to analyze expression of *CaHSP18.1a* in different tissues. For the thermotolerance treatment, R9 and B6 pepper seedlings (again, at the 4-to-6-leaf stage) were grown at 42 °C for 24 h, and root, stem, and leaf samples were collected from stress-treated seedlings at 0, 0.5, 1, 3, and 6 h post-treatment. For the drought stress treatment, the roots of R9 seedlings were soaked in 300 mM mannitol, and root, stem, and leaf samples were collected from stress-treated seedlings at 0, 3, 6, 12, and 24 h after treatment.

To analyze the function of *CaHSP18.1a* in response to pepper abiotic stress, silenced pepper seedlings and TRV2:00 pepper seedlings were grown at 42 °C for 24 h. For the drought and salt stresses, seedlings were treated with 300 mM mannitol and 300 mM NaCl for 24 h. Samples were collected and malondialdehyde (MDA) content, total chlorophyll content, and relative electrolyte leakage (REL) were determined. To identify the tolerance of *CaHSP18.1a*-overexpression in *Arabidopsis thaliana* in response to heat, salt, and drought stress, T3-generation *Arabidopsis thaliana* and wild-type lines were treated as described.

For heat stress, 2-week-old OE3 seedlings were treated at 42 °C for 24 h. For drought stress, water was withheld from 3-week-old transgenic *Arabidopsis* seedlings for 10 d. Samples were collected to measure the total chlorophyll contents, MDA content, REL, the activity levels of CAT, SOD, and ascorbic acid peroxidase (APX). For salt stress, the seeds of WT and transgenic lines were sown on MS medium with 0, 100, and 150 mM NaCl, and the roots lengths were measured after 10 days of treatment. The germination rate was determined after 6 d. The 3-week-old WT and transgenic plants were irrigated with 200 mM NaCl solution for 7 days, once every 2 days.

## 2.7. Measurement of Physiological Indicators

REL was estimated using the thiobarbituric acid reaction [43]. Total chlorophyll content was determined in the leaves according to methods previously described by Arkus et al [44]. Lipid peroxidation was determined by measuring the MDA content following the method of Campos et al [45]. POD and SOD activity levels were measured following the methods of Guo et al [46]. APX activity was measured using the methods of Nakano and Asada [47]. CAT activity was determined following AebiH [48].

## 2.8. Statistical Analyses

The experimental data were analyzed using Origin (Origin Lab, Northampton, MA, USA) and SPPS (SPSS Inc., Chicago, IL, USA). Significance tests for differences between control and stress treatments were assessed at a $p \leq 0.05$ level of significance. All experiments were performed and analyzed separately based on three biological replicates.

## 3. Results

### 3.1. Expression of the CaHSP18.1a in Pepper Plants under Abiotic Stress

To confirm whether heat, drought, and salt have an effect on the expression of *CaHSP18.1a*, R9 and B6 pepper lines were used to analyze the expression of *CaHSP18.1a* under heat, drought, and salt stress. Under heat stress (Figure 1A,B), the expression levels of *CaHSP18.1a* were significantly upregulated (samples IV, V, and VI) at 3 h in the roots, stems, and leaves of R9 plants and peaked at 6 h (sample Point V) in B6 plants. However, during the 22 °C recovery stage, the recovery times of *CaHSP18.1a* in the R9 and B6 strains differed. In roots, the expression levels of *CaHSP18.1a* in R9 and B6 plants returned to baseline after a 6 h recovery at the normal temperature (sample VII point) (Figure 1B). In stems, the expression level of *CaHSP18.1a* returned to a normal level after 3 h under the 22 °C recovery conditions for both the R9 and B6 plants (point VI in samples) (Figure 1C). In leaves, the expression level of *CaHSP18.1a* returned to normal levels at 24 h (sample VIII point) (Figure 1D). We also analyzed the expression pattern of *CaHSP18.1a* under salt and drought stresses (Figure 1E–H). After 6-h NaCl treatments at different concentrations, the expression of *CaHSP18.1a* in R9 and B6 leaves and roots was highest under the 150 mM NaCl, 100 mM NaCl treatments, respectively. The transcription of *CaHSP18.1a* was higher in R9 under different concentrations of NaCl treatment (Figure 1E, F). After 6-h treatments with different concentrations of mannitol, the expression of *CaHSP18.1a* in R9 and B6 leaves was the highest after the 150 mM mannitol treatment; the highest expression of *CaHSP18.1a* was observed in R9 and B6 roots subjected to the 50mM mannitol treatment (Figure 1G, H). In addition, the transcription of *CaHSP18.1a* was higher in R9 under different concentrations of mannitol. This analysis showed that the expression of *CaHSP18.1a* in pepper was induced by heat, salt, and drought. The response times of *CaHSP18.1a* in different organs of different pepper lines differed, and the response was more intense and more early in the R9 line, which suggests that *CaHSP18.1a* plays a substantial role in plant responses to heat stress.

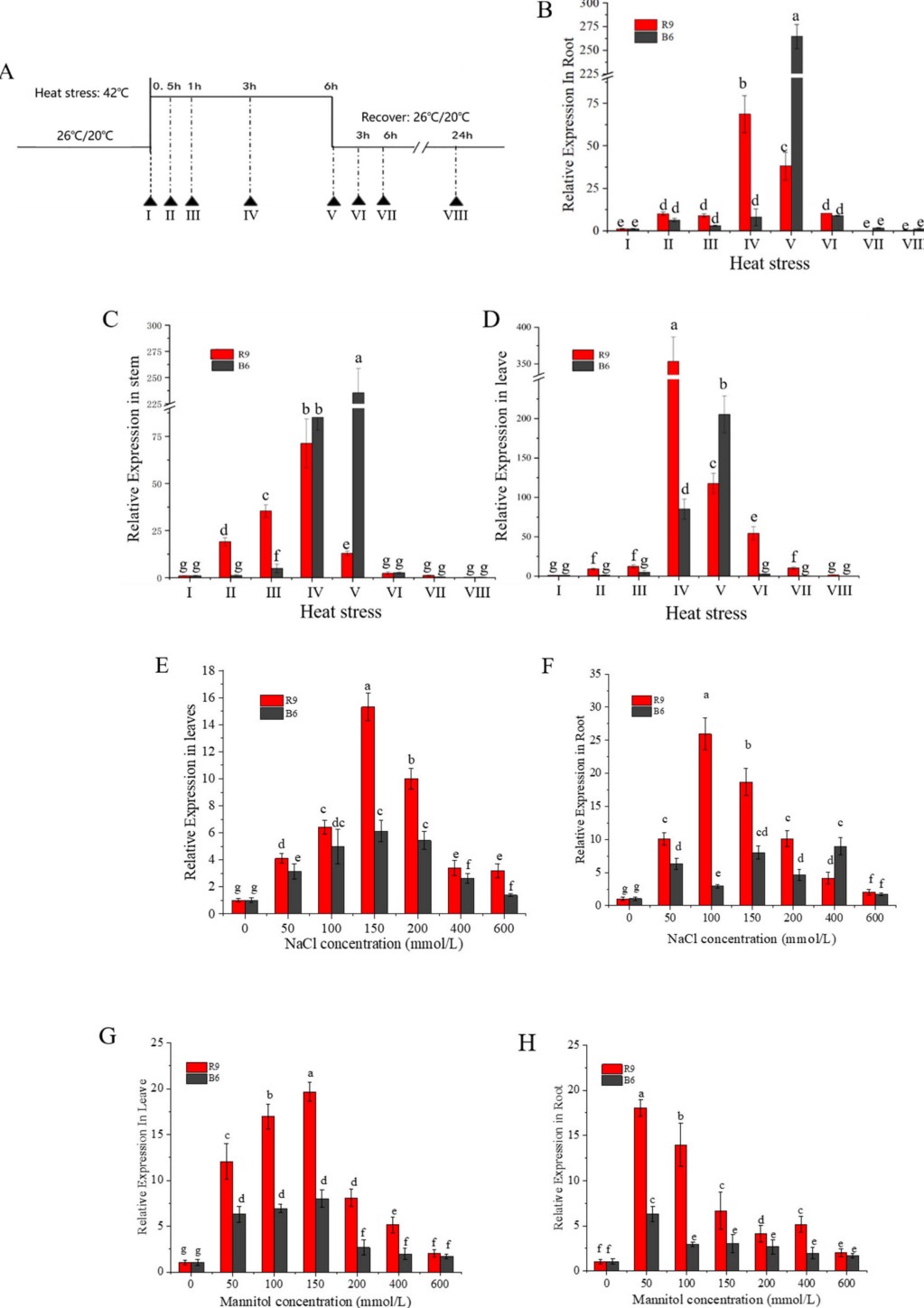

**Figure 1.** The expression characteristics of *CaHSP18.1a* in peppers in response to heat stress. (**A**) Time course of heat stress treatment and normal temperature recovery; the sampling time points are represented by triangles (pepper sample points I–VIII); (**B**–**D**) The expression levels of *CaHSP18.1a* in roots, stems, and leaves of R9 and B6 plants at each sampling time point; the expression levels of B6 and R9 plants were based on the reference level of their samples, and *CaUBI-3* was selected as the reference gene.(**E**–**H**) The expression levels of *CaHSP18.1a* following salt and drought treatment in R9 and B6 leaves and roots. The data presented are means with standard deviations of three biological replicates. Different letters denote statistical significance ($p \leq 0.05$).

### 3.2. Subcellular Localization of CaHSP18.1a Protein

To explore the subcellular localization of CaHSP18.1a, we constructed the pVBG2307: CaHSP18.1a: GFP fusion expression vector. Both pVBG2307: GFP and pVBG2307:CaHSP 18.1a: GFP fusion plasmids were introduced into *Nicotiana tabacum* leaves, and fluorescence was confirmed in the transformed tobacco cells with a microscope (Figure 2). We found that the green fluorescence signal of pVBG2307: CaHSP18.1a: GFP was detected in the cell membrane (Figure 2A), while the fluorescence of the empty pVBG2307: GFP vector was distributed throughout the cell (Figure 2B), indicating that CaHSP18.1a is localized to the cell membrane.

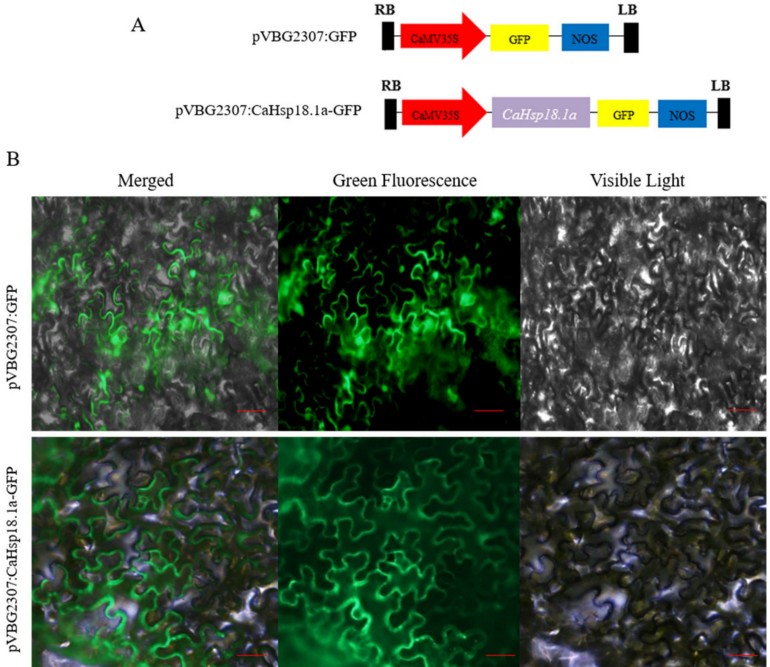

**Figure 2.** Transient expression of CaHSP18.1a in tobacco. (**A**) Schematic diagram of the CaHSP18.1a subcellular localization expression vector. (**B**) Subcellular localization of the CaHSP18.1a protein in tobacco leaves, with pVBG2307: GFP as control. Scale bar = 50 μm.

### 3.3. CaHSP18.1a-Silenced Plants Sensitive to Abiotic Stress

Confirming the VIGS procedure, after about 40–45 days, plants injected with the positive control TRV2: *CaPDS* showed a large area of typical white leaves (Supplementary Figure S1A), while under normal conditions there was no difference between *CaHSP18.1a*-silenced (TRV2:*CaHSP18.1a*) and negative control (TRV2:00) pepper plants. The silencing efficiency of *CaHSP18.1a*-silenced and TRV2:00 plants was assessed using q RT-PCR. As shown in Supplementary Figure S1A, the expression level of *CaHSP18.1a* in the silenced pepper plants decreased to less than 20% of that observed in the negative control plants. Thus, the silencing efficiency for *CaHSP18.1a*-silenced plants reached more than 80% (Supplementary Figure S1B). Therefore, control plants (TRV2:00) and silenced plants (TRV2:*CaHSP18.1a*) were used for the follow-up investigation.

HS (42 °C) was applied to *CaHSP18.1a*-silenced and control pepper plants for 3 h, and the silenced plants and the control group began to show different degrees of wilting. The heat-stress treatment (42 °C) induced significantly different symptoms after 24 h, such that the new growth of *CaHSP18.1a*-silenced plants was seriously wilted with curled leaves and shed lower leaves, while the leaves of the control plants were only slightly curled (Figure 3A). In addition, the MDA content and REL was lower in the control plants compared to *CaHSP18.1a*-silenced pepper plants (Figure 3B,C); however, the total chlorophyll content was higher in the control than in the *CaHSP18.1a*-silenced plants (Supplementary Figure S1C).

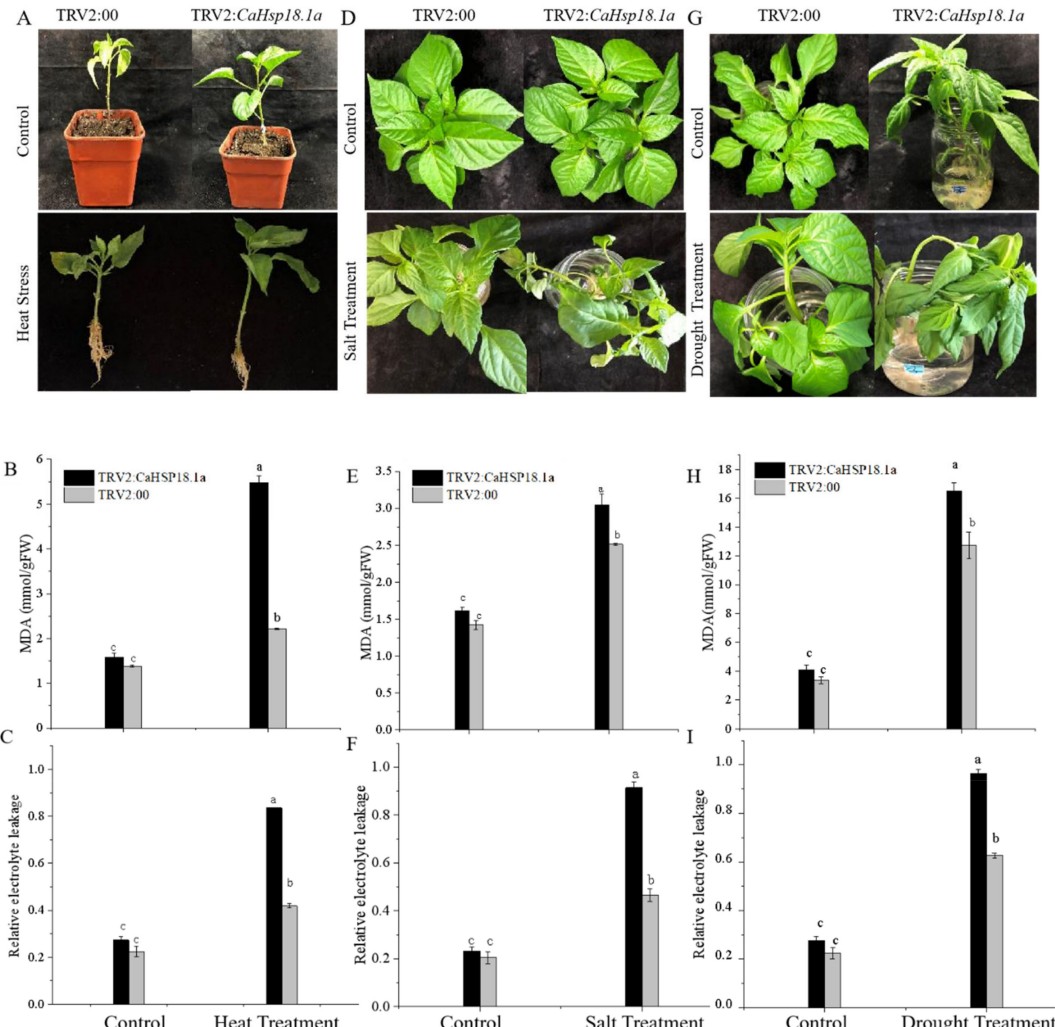

**Figure 3.** TRV2:*CaHSP18.1a* and TRV2:00 plant phenotypes under heat, drought, and salt treatments, respectively. (**A–C**) TRV2:*CaHSP18.1a* and TRV2:00 plant phenotypes, malonaldehyde (MDA) content, and relative electrolyte leakage (REL) under the 42 °C heat treatment for 24 h; (**D–F**) TRV2:*CaHSP18.1a* and TRV2:00 plant phenotypes, MDA content, and relative electrolyte leakage (REL) following salt stress by being soaked in 300 mM NaCl solution for 24 h; (**G–I**) TRV2:*CaHSP18.1a* and TRV2:00 plant phenotypes, MDA content, and REL following drought stress by being soaked in 300 mM mannitol solution for 24 h. Data are means with standard deviations of three biological replicates. Different letters denote statistical significance ($p \leq 0.05$).

To study the salt-tolerance of silenced and control plants, we washed their roots and soaked them in 300 mM NaCl solution for 24 h. The leaves of silenced plants showed symptoms of wilting, shriveling, and serious yellowing, with lower leaves that had begun to absciss, while the leaves of the control plants only showed some yellowing and did not exhibit obvious wilting. The leaves of the control plants showed only yellowing and no apparent wilting (Figure 3D). The MDA content of both plants increased significantly, but that of silenced plants was higher than that of control plants (Figure 3E). Relative electrolyte leakage (REL) was higher in silenced plants compared to control plants (0.96 versus 0.65) (Figure 3F). To study the effects of *CaHSP18.1a*-silencing on drought tolerance, the silenced and control plants were soaked in 300 mM mannitol solution for 36 h. The *CaHSP18.1a*-silenced pepper showed severe loss of water and wilting, while control plants showed no obvious change (Figure 3G). Furthermore, the MDA content and REL both exhibited a similar increase in the silenced pepper plants (Figure 3H,I). This indicated that silencing of *CaHSP18.1a* reduced the drought tolerance of pepper plants.

### 3.4. Effect of CaHSP18.1a Overexpression on Transgenic Arabidopsis

3.4.1. Overexpression of CaHSP18.1a Enhances Plant Tolerance of Heat Stress

First, we transformed pVBG2307:*CaHSP18.1a* into *Agrobacterium tumefaciens* strain GV3101, which was used to transfect *Arabidopsis thaliana* using the dipping method; successful transformants were identified through resistance gene screening and molecular level detection until homozygous T3 lines were obtained (Supplementary Figure S2A). The wild-type (WT) line and five transgenic lines were cultured on Murashige and Skoog (MS) medium for 10 days, and the lengths of their roots were measured. The survival rate of WT plants was lower than those of the OE1, OE2, and OE3 seedlings (Supplementary Figure S2C,D). Transgenic plants and WT plants were cultured under normal growth conditions for 48 days, and the growth rates of the OE1, OE2, and OE3 lines exceeded those of WT plants (Supplementary Figure S2D). Thus, the OE3, OE2, and OE1 lines were selected for follow-up experiments.

The obtained transgenic lines and WT plants were heat treated (42 °C for 24 h) at the 3-week stage. After heat-stress treatment, the WT plants showed wilting symptoms. Notably, restorable wilt or indistinct-symptoms were observed among the *CaHSP18.1a*-OE seedlings (Figure 4A), indicating that *CaHSP18.1a* plays an active role in increasing the thermotolerance of transgenic *Arabidopsis*. In addition, REL and MDA content increased significantly in both the OE and WT lines after heat treatment, while the MDA content was notably lower in the transgenic lines relative to the WT plants (Figure 4B). In addition, the SOD and POD activities of *CaHSP18.1a*-OE seedlings were clearly higher than those of the WT plants (Figure 4C,D). However, the catalase (CAT) activity did not significantly differ between the WT and transgenic lines (Figure 4E). *CaHSP18.1a* played a role in transgenic *Arabidopsis*, probably by regulating the expression of endogenous genes.

In the present study, among 18 stress-related genes, 12 were up-regulated in transgenic lines, while the other 6 did not change much (Figure 5). Among the up-regulated genes, *AtHSPC30, AtAPX3, AtCAT, AtHSP70*, and *AtRab1* were more prominently expressed in the transgenic OE3 line. Moreover, the expression of the 18 stress-related genes was markedly increased in both transgenic *Arabidopsis* and WT plants under heat stress. However, the expression of these genes in WT seedlings was lower than that in transgenic seedlings (Figure 5).

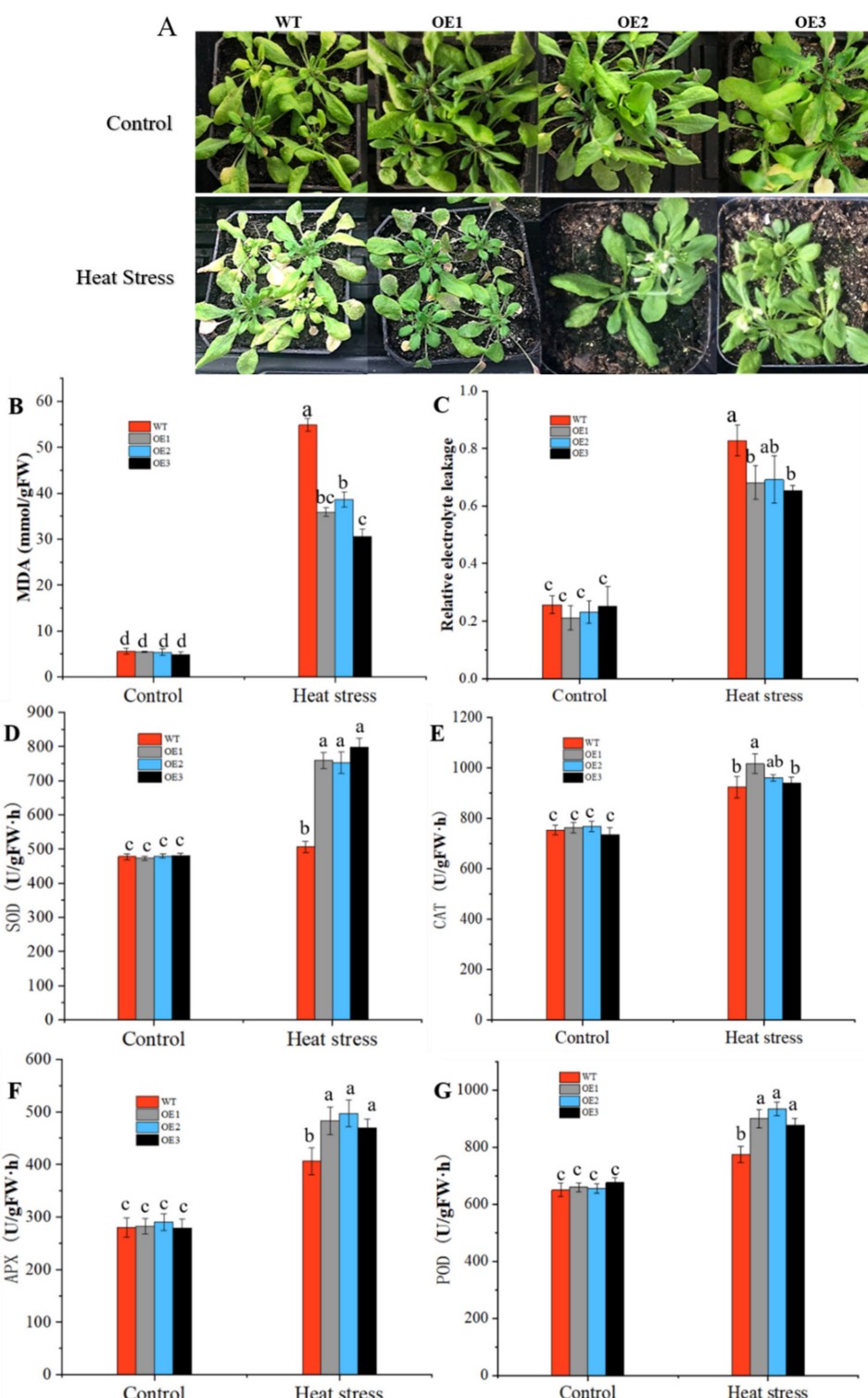

**Figure 4.** Heat resistance of transgenic *CaHSP18.1a*-OE *Arabidopsis* plants. (**A**) Phenotypes of 42 °C-treated wild-type (WT) and transgenic *Arabidopsis*; (**B–C**) Malonaldehyde (MDA) and relative electrolyte leakage (REL) of WT and transgenic *Arabidopsis*; (**D–G**) Superoxide dismutase (SOD), catalase (CAT), peroxidase (POD), and ascorbic acid peroxidase (APX) activity of WT and transgenic *Arabidopsis*. Data are means with standard deviations of three biological replicates. Different letters denote statistical significance ($p \leq 0.05$).

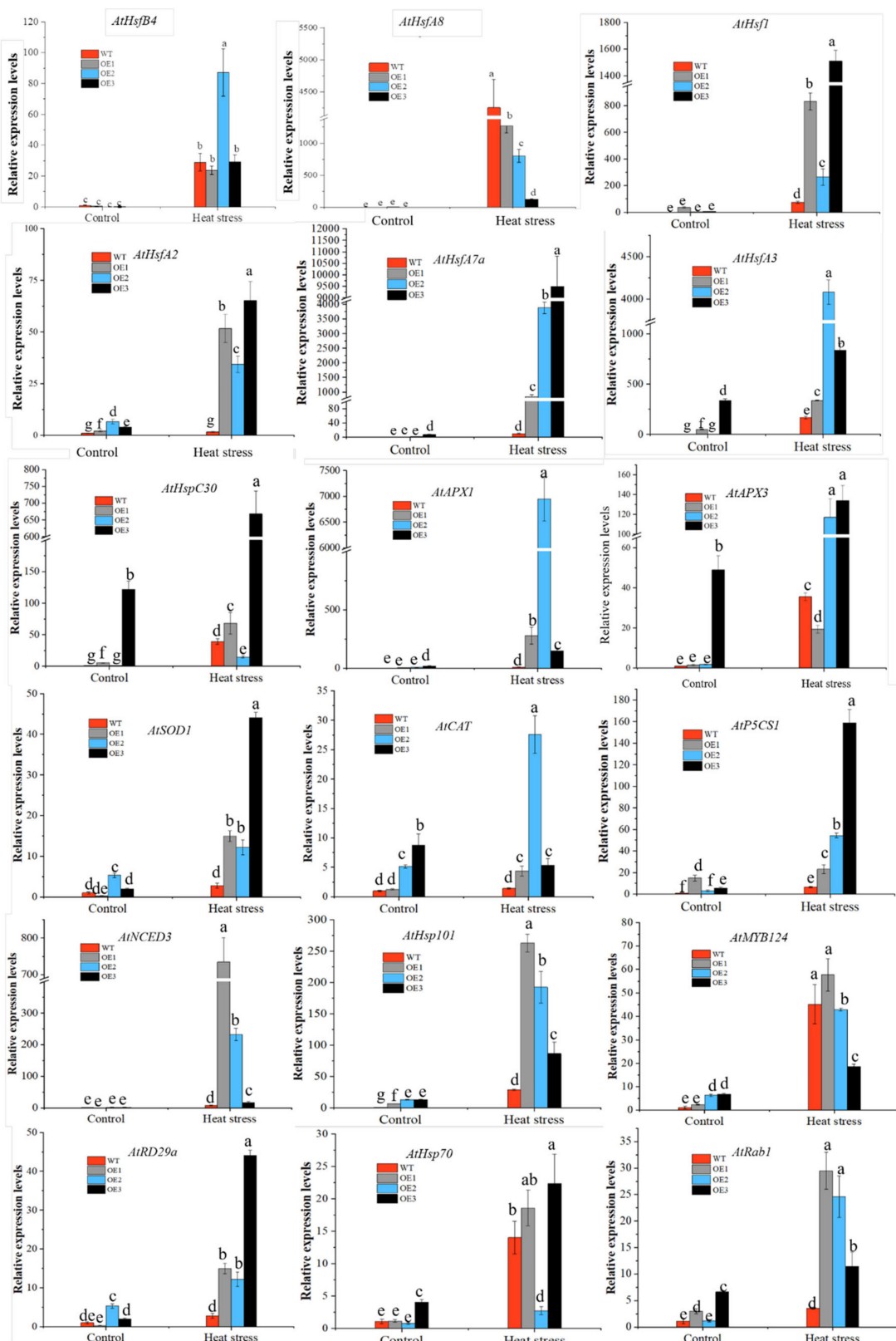

**Figure 5.** Expression patterns of stress–response genes in wild-type (WT), OE1, OE2, and OE3 lines before and after 42 °C heat treatment for 24 h. Data are means with standard deviations of three biological replicates. Different letters denote statistical significance ($p \leq 0.05$).

### 3.4.2. Overexpression of CaHSP18.1a Enhances Plant Tolerance to Drought Stress

CaHSP18.1a is a molecular chaperone, though its response to drought stress is still unclear. To further study its function under drought and salt stress, *CaHSP18.1a* transgenic *Arabidopsis thaliana* and WT seedlings were drought treated (Figure 6A). After water control treatment was conducted for 10 d on 3-week-old plants with consistent growth, WT plants showed severe wilting; the leaves turned yellow, while overexpression plants grew better than WT plants (Figure 6A). These results indicated that *CaHSP18.1a* increases drought tolerance of transgenic *Arabidopsis*. In addition, the MDA content and REL were increased in both the WT and OE lines, whereas the MDA content of transgenic seedlings was obviously lower than that of WT plants (Figure 6B,C). Thus, the degree of damage in OE plants was lower than that in WT plants. The SOD, CAT, and peroxidase (POD) activity showed an upward trend in both the WT and OE lines, but the activity level of SOD, CAT, and POD in transgenic seedlings was notably higher than that in WT plants (Figure 6D,F,G). While ascorbic acid peroxidase (APX) activity increased, there was, however, no visible difference between *CaHSP18.1a*-OE and WT lines (Figure 6E). The expression levels of the 18 stress-related genes were induced to varying degrees by drought stress. However, the expression of *AtHsfA2*, *AtHSPC30*, and *AtAPX1* exhibited almost no change in WT seedlings, and all of them were strongly increased in *CaHSP18.1a*-OE lines after drought stress. In addition, the expression levels of other genes were higher in transgenic lines compared to the WT plants after drought stress (Figure 7). Thus, the 18 stress-related genes examined may be involved at different levels in the response of *CaHSP18.1a*-OE lines to drought stress.

### 3.4.3. Overexpression of CaHSP18.1a Enhances Plant Tolerance to Salt Stress

To study the role of *CaHSP18.1a* in salt stress, transgenic *Arabidopsis thaliana* was subjected to salt stress. First, the germination rate of transgenic seeds under salt stress was observed (Supplementary Figure S3A). OE2, OE1, and WT seeds exhibited normal germination on MS plates without NaCl; the germination rate and seedling growth of transgenic lines were almost unaffected by 100 mM NaCl MS plates. However, the seed germination rate differed when the NaCl concentration was increased to 150 mM, and the germination rate of WT seeds was slightly lower than that of transgenic seeds. The salt tolerance of *CaHSP18.1a* transgenic *Arabidopsis* seeds increased. After the germinated seedlings were moved to MS plates with NaCl concentrations of 0, 100, and 150 mM, compared with the untreated seedlings, the growth of OE2, OE3, OE1, and WT seedlings after 6 d under 100 and 150 mM NaCl treatments was worse (Supplementary Figure S3A,B); root elongation was significantly inhibited, and the root length of transgenic lines seedlings was greater than that of WT plants (Figure 8A).

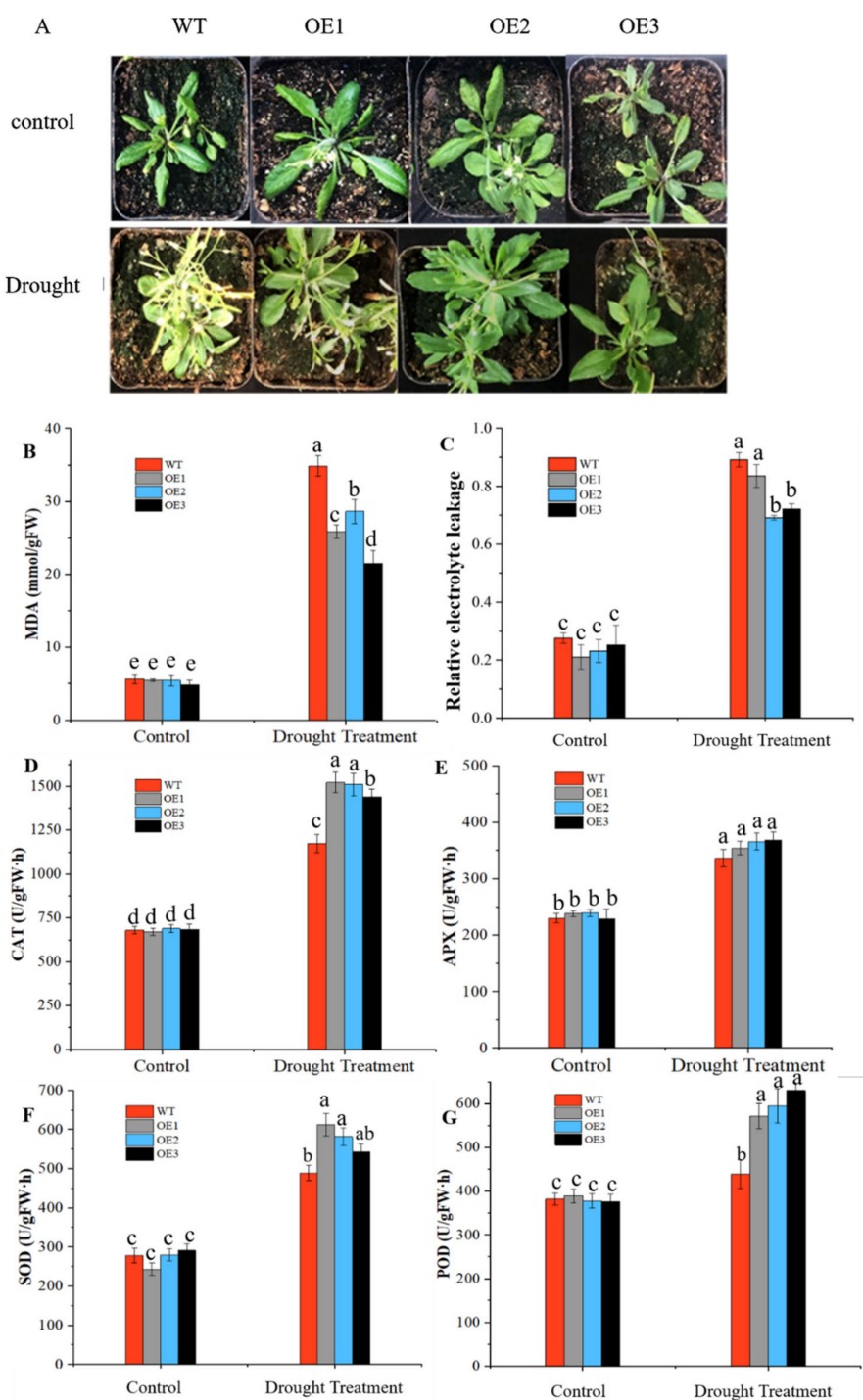

**Figure 6.** Drought resistance of transgenic *CaHSP18.1a*-OE *Arabidopsis* plants. (**A**) Phenotypes of wild-type (WT) and *CaHSP18.1a*-OE *Arabidopsis*; (**B,C**) Malonaldehyde (MDA) content and relative electrolyte leakage (REL) of WT and *CaHSP18.1a*-OE *Arabidopsis*; (**D–G**) Superoxide dismutase (SOD), catalase (CAT), peroxidase (POD), and ascorbic acid peroxidase (APX) activity of WT and *CaHSP18.1a*-OE *Arabidopsis* without watering for 10 days. Data are means with standard deviations of three biological replicates. Different letters denote statistical significance ($p \leq 0.05$).

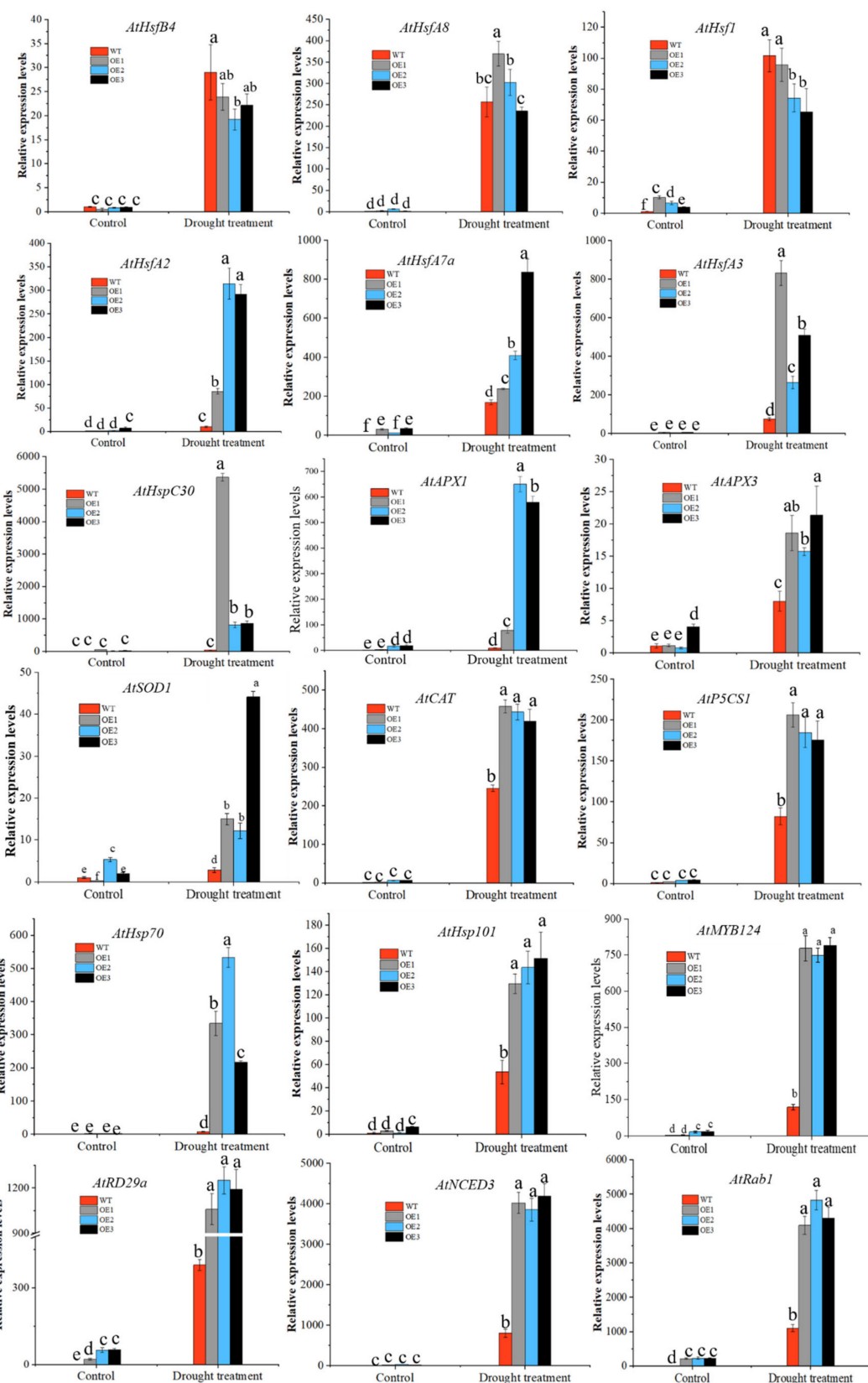

**Figure 7.** Expression pattern of stress–response genes in wild-type (WT), OE1, OE2, and OE3 lines before and after 10 days of drought treatment. Data are means with standard deviations of three biological replicates. Different letters denote statistical significance ($p \leq 0.05$).

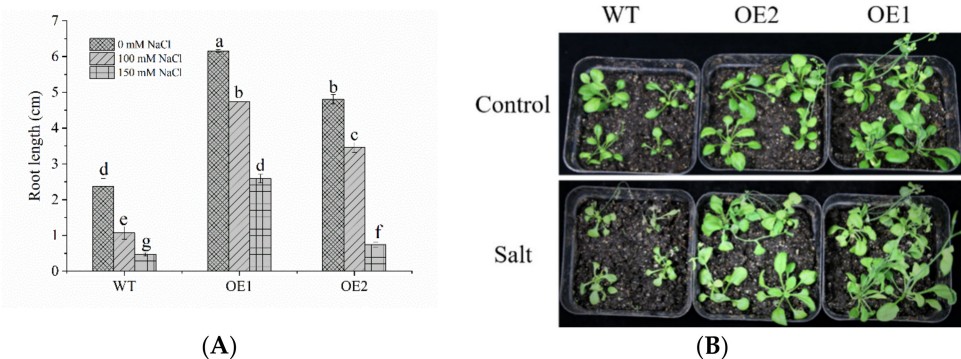

**(A)**                    **(B)**

**Figure 8.** Overexpression of *CaHSP18.1a* enhanced tolerance to salt stress. (**A**) Root length of wild-type (WT) and *CaHSP18.1a*-OE *Arabidopsis* lines grown for 10 d on Murashige and Skoog (MS) medium containing 0, 100, and 150 mM NaCl. (**B**) Seedling growth for WT and transgenic plants exposed to NaCl. Three-week-old seedlings were watered with 200 mM NaCl once every 1–2 days. The images were taken after 7 days.

To explore the effect of *CaHSP18.1a* on the salt tolerance of *Arabidopsis thaliana*, 3-week-old WT and transgenic plants were irrigated with 200 mM NaCl solution for 7 days. As shown in Figure 8B, WT plants showed dehydration and wilting with weak growth. However, the transgenic *Arabidopsis thaliana* showed no significant difference from the control plants except for a slight yellowing phenotype in their leaves.

## 4. Discussion

Plants are inevitably subjected to various extreme environmental conditions, such as heat, drought, oxidation and salt damage [36]. Under such adverse conditions, sHsP20s make a valuable protective contribution [19]. Plant HSPs are linked to heat tolerance and have been confirmed in many species [31,49]. Guo et al. [30] identified sHsP20s in pepper, and showed that *CaHSP18.1a* was induced in different tissues of pepper plants under heat stress, but the function of *CaHSP18.1a* under heat, salt, and drought stress has not been further studied. In this study, we identified that *CaHSP18.1a* is positively involved in plant tolerance to heat and salt, drought stress.

*CaHSP18.1a* was responsive to heat stress in both R9 and B6 plants and strongly induced [30]. In accordance with the results, we also found that the expression level of *CaHSP18.1a* was strongly induced in both the R9 and B6 lines after heat stress treatment (Figure 1B–D). However, under heat stress, the expression of *CaHSP18.1a* in R9 plants was higher than that in B6 plants (Figure 1B–D). This may be because R9 is a thermo-tolerant cultivar, it has better thermo-tolerance and adaptability than B6 under heat stress. The heat tolerance of plants is related to the dynamic expression patterns of heat stress-related genes [50]. Under heat stress, other HSP20s or HSPs in R9 are also strongly and rapidly induced in the early stage of heat stress (0.5–1 h). It had also been reported that the expression level of *CaHSP25.8* and *CaHSP30.1* in R9 was higher than B6, but with the extension of heat stress treatment time, the expression level of these two genes in B6 were higher than R9 [30]. However, these results also showed that the expression of *CaHSP18.1a* was lower at V in R9 than in B6. The expression of pepper HSP20s is regulated by many transcription factors, such as HSFs [51]. Under heat stress, it is because HSFs that regulate the expression of *CaHSP18.1a* in pepper variety R9 and B6 are different, or the expression of the same HSFs that regulate the expression of *CaHSP18.1a* is different in R9 and B6, causing the differential expression of *CaHSP18.1a* in B6 and R9 [51,52]. Therefore, the difference in the expression of *CaHSP18.1a* between heat-resistant and heat-sensitive varieties is due to the above reasons. However, the relationship between the function of this gene and the heat-resistance mechanism of pepper still needs further research.

In addition, *CaHSP18.1a* was induced under salt and drought stress (Figure 1E–H). The expression of *CaHSP18.1a* in R9 leaves and roots was highest under the 150 mM, 100 mM NaCl treatments, respectively (Figure 1E,F). The expression of *CaHSP18.1a* in R9

leaves was the highest after the 150 mM mannitol treatment; the highest expression of *CaHSP18.1a* was observed in R9 roots subjected to the 50 mM mannitol treatment (Figure 1G,H). However, the expression of the *CaHSP18.1a* decreased at higher NaCl and mannitol concentrations. The response pattern of *CaHSP18.1a* that rapidly and sharply responded to salt and drought stress in a short time, and then had slight variations, was similar to quite a few HSP20s such as *TaHSP23.9* [53] and *ClHSP22.8* [54]. Thus, *CaHSP18.1a* may play a role in pepper which rapidly adapts to drought and salt stress.

VIGS technology is an important method used to study gene function under adverse environments [55]. In the R9 line, silencing of *CaHSP16.4* reduces heat tolerance and drought resistance of pepper plants [31]; *CaHSP22.0*-silenced peppers showed more sensitivity to salt and heat stress, which was mainly reflected in decreased antioxidant enzyme activity, increased leaf conductivity, and increased superoxide anion and MDA contents [34]. MDA content and REL are products of cell membrane lipid peroxidation, which damages the integrity of plasma membranes under salt or heat stress [35] and may sensitize plants to subsequent stress [56]. MDA content, total chlorophyll content, and REL are widely used to determine the degree to which plants have been damaged by abiotic stress [4]. It has also been reported that proline content, MDA content, and POD and SOD activity of pepper were significantly related to the variation in heat tolerance and temperature stress time, which can be used as an index for heat resistance identification [57].

In this study, after treatments with high temperature, salt, and drought stress, the content of MDA and REL in peppers that had been silenced for *CaHSP18.1a* was higher than that of the controls, indicating that the damage to cell membranes increased in *CaHSP18.1a*-silenced plants (Figure 3A–C). It was also found that the *CaHSP18.1a*-silenced plants had lower total chlorophyll content when exposed to heat stress (Figure 3B and Supplementary Figure S1C). These results demonstrated that silencing of *CaHSP18.1a* reduced pepper stress tolerance (Figure 3). In contrast, overexpression of *CaHSP18.1a* in *Arabidopsis thaliana* transgenic lines was associated with minimal injury symptoms, increased REL, and decreased MDA content compared with WT plants (Figures 4 and 6). These results showed that *CaHSP18.1a* increases plant tolerance to heat, salt, and drought stresses.

HSP20s are widely distributed in plants, and their location may be related to their function, as exemplified by *AtHsP21* being localized to chloroplasts [23]. *CaHSP18.1a* was predicted to have cytoplasm localization [30]. Subcellular localization of *CaHSP18.1a* is shown in Figure 2, which confirmed that it is localized to the cell membrane.

Studies have shown that HSP20 is a molecular chaperone that can also participate in antioxidant mechanisms of plants [6,58]. HSP20scan cooperate with the plant's antioxidant scavenging system to protect plants from secondary damage [59,60]. For example, overexpression of *AtHSP17.6* can increase CAT enzyme activity and further regulate abiotic stress responses [61]. Furthermore, the over-expression of *ZmHSP16.9* in tobacco can increase the activities of POD, CAT and SOD, and enhance oxidative stress tolerance [62]. In this study, SOD, CAT, POD, and APX enzyme activities in overexpression of *CaHSP18.1a* plants were significantly enhanced under heat and drought stress. This is similar to the results obtained with *Arabidopsis* transformed with *CaHSP25.9*; that is, by increasing the activities of ROS-scavenging related antioxidant enzymes, the heat, salt, and drought tolerance of plants can be increased [31,63]. It has been reported that plants have built defense mechanisms that scavenge excess reactive oxygen species (ROS) throughout their long evolutionary histories [64–66], such as ROS-scavenging non-enzymatic antioxidants (e.g., ascorbic acid (AsA), glutathione, and proline) [67] and antioxidant enzymes (e.g., peroxidase (POD), catalase (CAT), superoxide dismutase (SOD), and glutathione peroxidase (GPX)) that prevent secondary oxidative stress caused by abiotic stress [68–70]. Moreover, the expression levels of *AtSOD1, AtAPX1, AtAPX3*, and *AtCAT1* were also increased by heat and drought stress, and were higher in the *CaHSP18.1a* transgenic *Arabidopsis* than in the WT. This indicated that *CaHSP18.1a* may improve stress resistance through the ROS-scavenging system, but the specific mechanism needs further study.

Many stress-related genes are involved in plant responses to heat, salt, and drought stresses. It has been reported that *AtHsfA2* is a heat shock transcription factor that enables prolonged acquired thermo-tolerance, and it can enhance tolerance to salt and osmotic stresses [71–73]. s HSPs are downstream target genes of HsfA2 [74]. Burke [75,76] have also shown that *AtHSA32* and *AtHSP101* expression can be induced by high temperature and participate in the acquired thermo-tolerance of plants. *AtMYB44* can be induced by salt, drought, and other stresses to participate in the abscisic acid (ABA) signaling pathway; Refs. [77,78] found that the ABA signaling response gene *AtDREB2A* can be induced by low temperature stress. The drought responsive gene *AtRD29A* was up-regulated under heat, salt, and drought stresses [79], while the molecular chaperone HSP70 participates in drought and heat stress responses [28]. *NCED3* is related to biological metabolism and also participates in defense responses to drought stress [80]. Notably, HSP20s can regulate many of these stress-related genes [31,36]. For example, *CaHSP16.4* and *OsMSR-4* can increase the expression of these genes in transgenic seedlings, thereby enhancing stress resistance [31,81]. In this study, we assessed the expression levels of 13 stress-related genes in WT and transgenic plants. *CaHSP18.1a* enhanced heat tolerance in transgenic *Arabidopsis*, which may be closely related to its regulation of the expression of many heat-stress-related genes in *Arabidopsis*. Overexpressed genotypes compared with wild type under normal conditions also showed higher values of stress-related genes expression, higher expression of *AtP5CS,* higher expression of *AtNCED*, higher expression of *AtMYB*, and higher expression of *AtRD29, AtHsfA2, AtRab1* and *AtHSP30*. These results showed that *CaHSP18.1a* may play an important role in regulation of these genes. Under heat stress, the expressions levels of *AtHsfB4, AtHSFA8, AtHSFA2, AtHSFA7a, AtHSPC30, AtHSFA3, AtHSP70,* and *AtHSP101* in transgenic plants were significantly higher than WT plants (Figure 5). In particular, the *AtHSPC30* and *AtHSP70* transcripts were present at levels nearly 3-fold higher in transgenic seedings than in WT plants; the transcript level of *AtHsfA2* was also up-regulated in transgenic plants. The expression levels of *AtHSP70, AtHSP101, AtDREB2A, AtMYB124, AtNCED3, AtRD29A,* and *AtRAB1* were higher (Figure 7) in the *CaHSP18.1a* transgenic *Arabidopsis* than in WT plants under drought stress. Similar results were also reported by Feng and Huang [31,32]. Thus, *CaHSP18.1a* may respond to heat and drought stress through its complex regulatory network.

## 5. Conclusions

In this study, we first analyzed the expression of *CaHSP18.1a* in R9 and B6 pepper lines and demonstrated that *CaHSP18.1a* was expressed when induced by abiotic stress factors such as high temperature, drought, and high salinity. *CaHSP18.1a* silencing decreased the resistance of pepper plants to heat, drought, and salt stresses through different molecular and physiological mechanisms. Overexpression analyses of *CaHSP18.1a* in transgenic *Arabidopsis* further confirmed that *CaHSP18.1a* functions positively in responses to heat, drought, and salt stresses. The expression levels of other stress-related genes were also measured, and some were determined to be significantly affected by *CaHSP18.1a* overexpression. We further confirmed that *CaHSP18.1a* protein was localized in the cell membrane. Collectively, these results show *CaHSP18.1a* likely acts as a positive regulator of the response to abiotic stresses in pepper.

**Supplementary Materials:** The following are available online at https://www.mdpi.com/article/10.3390/horticulturae7050117/s1, Figure S1: Detection of silencing efficiency of CaHsp18.1a gene mediated by TRV2; Figure S2: Validation and acquisition of homozygous strain of T3 generation of Arabidopsis with overexpression of CaHsp18.1a; Figure S3: Germination of the transgenic Arabidopsis under salt stress; Table S1: The main primers sequence used in this research.

**Author Contributions:** S.L., Y.-L.L. and Z.-H.G. conceived and designed the research; S.L. and Y.-L.L. conducted the experiments and wrote the manuscript; G.-X.C. and Y.-L.L. analyzed the data; H.S.u., S.L. and J.-J.X. critically revised the manuscript; Z.-H.G. contributed reagents and funded the project. All authors have read and agreed to the published version of the manuscript.

**Funding:** We highly appreciate the financial support of the funding from the National Natural Science Foundation of China (No. U1603102, No. 31772309).

**Institutional Review Board Statement:** Not applicable.

**Informed Consent Statement:** Not applicable.

**Data Availability Statement:** Data is contained within the article.

**Conflicts of Interest:** The authors declare no conflict of interest.

## Abbreviations

| | |
|---|---|
| ACD | alpha-crystallin domain |
| HS | heat stress |
| HSP | heat shock proteins |
| sHSPs | small heat shock proteins |
| REL | relative electrolyte leakage |
| MDA | Malondialdehyde |
| OE | Overexpression |
| OE1 | No. 1 Arabidopsis line with overexpressed CaHSP18.1a |
| OE2 | No. 2 Arabidopsis line with overexpressed CaHSP18.1a |
| OE3 | No. 3 Arabidopsis line with overexpressed CaHSP18.1a |
| R9 | a thermo-tolerant line |
| qRT-PCR | real-time fluorescence quantitative PCR |
| VIGS | virus-induced gene silencing |
| PDS | phytoene desaturase |
| TRV | tobacco rattle virus |
| ROS | reactive oxygen species |
| APX | ascorbate peroxidase |
| CAT | catalase |
| SOD | superoxide dismutase |
| POD | peroxidase |

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
