# Peer review of "CaHSP18.1a, a Small Heat Shock Protein from Pepper (Capsicum annuum L.), Positively Responds to Heat, Drought, and Salt Tolerance"

_horticulturae, doi:10.3390/horticulturae7050117_

Round 1
Reviewer 1 Report
Very interesting topic and remarkable experimental results. However, I find that this manuscript is not suitable for publication in its present form for the following reasons:
- According to the title, the manuscript deals with abiotic stress response in pepper plants and a set of different experimental results are reported (expression of CaHsp18.1° in pepper under abiotic stress, subcellular localization and silencing) but the section about overexpression of this gene deals with experiments made on Arabidopsis (not pepper). In my opinion these two different research lines should be reported in separate manuscripts.
- The Discussion section is not adequate, neither its content or its style.
Following are my additional comments.
Abstract
Mostly well written. Few spelling mistakes (e.g. missing “a” in “mechnism”, line 18) should be spell checked or mistypings like in “thatthe”, line 20. Some sentences may be rephrased (lines 17-18, or lines 28-29).
Introduction
Spell check and mistypings as above.
Lines 70-96 is too long and may be summarized
Results
Section 3.1 to 3.3 mostly OK. The section 2.2, between 3.1 and 3.3 in the results section should be re-numbered.
Section 3.4, from 3.4.1.to 3.4.3 (overexpression in Arabidopsis) in my opinion should not be part of this manuscript. Moreover the writing style and english language is not consistent with the previous sections and it should be improved. Vague terminology should be avoided e.g “was much lower” in line 340 and quantified (lines 346 “severe”, “slight” line 347, “increased significantly” line 349, etc …)
Discussion
Should be improved, shortened and irrelevant parts deleted.
Some parts (e.g.lines 464-477) are not relevant while other parts (lines 488-509) should be restructured to be more clear and more informative. There are long passages of text which may be more appropriate for the Introduction rather than Discussion section (lines 533-537, 538-544 and so on).
Overall, this section is excessively wordy, not much informative and I cannot see much discussion of the experimental results.
Author Response
Dear Editors,
We carefully read the comments and suggestions of you and revised our manuscript according to your comments. All the revisions in the manuscript are clearly highlighted using the "Track Changes" function in the Microsoft Word. The revised manuscript please see the attachment.
Our responses to the your comments are as follows:
Ponit 1: According to the title, the manuscript deals with abiotic stress response in pepper plants and a set of different experimental results are reported (expression of CaHsp18.1a in pepper under abiotic stress, subcellular localization and silencing) but the section about overexpression of this gene deals with experiments made on Arabidopsis (not pepper). In my opinion these two different research lines should be reported in separate manuscripts.
Response1: Thank you for the excellent suggestion. To validate the function of certain gene (S), is to silence or overexpress in a plant system and to observe the phenotypic changes along with studying the biochemical indices. Since, the overexpression in the system of pepper transgenic is not effective, the model plant Arabidopsis is a good choice. Here, the authors tried their level best to validate the hypothesis through heterologous expression following gene functional studies in the author's lab (Sun et al 2019, Feng et al 2019,)
Reference:
Feng, X.; Zhang, H.; Ali, M.; Gai, W.; Gong, Z., A small heat shock protein CaHsp25.9 positively regulates heat, salt, and drought stress tolerance in pepper (Capsicum annuum L.). Plant Physiology Biochemistry 2019, 142, 151-162.
Sun, J.; Cheng, G.; Huang, L.; Liu, S.; Ali, M.; Khan, A.; Yu, Q.; Yang, S.; Luo, D.; Gong, Z. Modified expression of a heat shock protein gene, CaHSP22.0, results in high sensitivity to heat and salt stress in pepper (Capsicum annuum L.). Scientia Horticulturae 2019, 249, 364-373.
Ponit2: The Discussion section is not adequate, neither its content or its style.
Response 2: Thanks for your comments; we have modified the discussion section in the revised manuscript.
Following are my additional comments.
Abstract
Ponit 3:Mostly well written. Few spelling mistakes (e.g. missing “a” in “mechnism”, line 18) should be spell checked or mistypings like in “thatthe”, line 20. Some sentences may be rephrased (lines 17-18, or lines 28-29).
Response 3: We have corrected the spelling mistakes in the revised manuscript (in “mechnism”, as “mechanism” line 18; “thatthe”, as “that the”line 20). In addition, we have rephrased the lines 17-18 and lines 28-29 in the revised manuscript.
Introduction
Ponit 4:Spell check and mistypings as above. Lines 70-96 is too long and may be summarized
Response 4 : We have organized, re-written, and summarized the lines 70-96 in the revised manuscript
Results
Ponit 5:Section 3.1 to 3.3 mostly OK. The section 2.2, between 3.1 and 3.3 in the results section should be re-numbered.
Response 5: Thank you, we have re-numbered the results section as 3.1, 3.2 and 3.3 in the revised manuscript.
Ponit 6:Section 3.4, from 3.4.1.to 3.4.3 (overexpression in Arabidopsis) in my opinion should not be part of this manuscript. Moreover the writing style and english language is not consistent with the previous sections and it should be improved. Vague terminology should be avoided e.g “was much lower” in line 340 and quantified (lines 346 “severe”, “slight” line 347, “increased significantly” line 349, etc …)
Response 6: Thank you, we have edited the language in the revised manuscript.
Discussion
Ponit 7:Should be improved, shortened and irrelevant parts deleted. Some parts (e.g.lines 464-477) are not relevant while other parts (lines 488-509) should be restructured to be more clear and more informative. There are long passages of text which may be more appropriate for the Introduction rather than Discussion section (lines 533-537, 538-544 and so on). Overall, this section is excessively wordy, not much informative and I cannot see much discussion of the experimental results.
Response 7: Thank you for the advice, we have deleted the irrelevant parts (e.g. lines 464-477), and other parts (lines 488-509) have restructured; lines 533-537, 538-544 were re-written in the revised manuscript.

Reviewer 2 Report
Dear authors,
This looks to be a well-conducted study and is well written. A few minor comments (please see PDF).
Introduction: a suitable background to the study, relevant literature cited. In the last paragraph, please clearly define the hypotheses and objectives of the study not your findings.
Materials and methods: The methodology looks to be suitable for such a study.
Results: are consistently presented.
Discussion: Quite comprehensive and with suitable references.
References: The list of literature is well-chosen.

Author Response
Dear Editors,
We carefully read the comments and suggestions of your's and revised our manuscript according to your comments. All the revisions in the manuscript are clearly highlighted using the "Track Changes" function in the Microsoft Word. The revised manuscript please see the attachment.
Our responses to the your comments are as follows:
This looks to be a well-conducted study and is well written. A few minor comments (please see PDF).
Ponit 1:Introduction: a suitable background to the study, relevant literature cited. In the last paragraph, please clearly define the hypotheses and objectives of the study not your findings.
Ponit 2:Materials and methods: The methodology looks to be suitable for such a study.
Ponit 3:Results: are consistently presented.
Ponit 4:Discussion: Quite comprehensive and with suitable references.
Ponit 5:References: The list of literature is well-chosen.
Response all: Thanks very much for the guidance for the improvement the article, we have modified and addressed all the minor comments in the revised manuscript.

Reviewer 3 Report
The article entitled “CaHsp18.1a, a small heat shock protein, responds positively heat and drought as well as salt tolerance in Capsicum annuum L. pepper" contains the results of interesting research and undoubtedly broadens the knowledge of plant heat shock proteins.
It is well known that the expression of HSPs increases when cells are exposed to stress factors, including elevated temperature, but also low temperature, salt, osmotic and heavy metals stress. The title is therefore a bit disappointing because it does not point a significant scientific achievement.
I did not find any errors in the work but I do not understand why the Authors did not decide to carry out additional tests on the pepper cultivar which is commonly planted.
The Authors did not clarify, using the abbreviation POD, whether they meant the sum of water soluble peroxidases (the ascorbate peroxidase is abbreviated properly).
I would also like to draw the Authors' attention to very small markers in some graphs, e.g. in Figs. 3 and 5, in the legends. They are almost invisible when reading the paper version.
Axis descriptions also have different sizes (please, compare the Fig. 3F and Fig. 3I).
The abreviation for a heat shock protein is written 'Hsp' and 'HSP' (e.g. in lines 75–77). This form should be unified.
There are many minor editing errors, e.g. missing spaces or extra spaces in the manuscript. The entire text needs to be corrected.
Author Response
Dear reviewer
We carefully read the comments and suggestions of , and revised our manuscript according to your comments. All the revisions in the manuscript are clearly highlighted using the "Track Changes" function in the Microsoft Word.
The revised manuscript please see the attachment.
Our responses to the comments are as follows:
Ponit 1:It is well known that the expression of HSPs increases when cells are exposed to stress factors, including elevated temperature, but also low temperature, salt, osmotic and heavy metals stress. The title is therefore a bit disappointing because it does not point a significant scientific achievement.
Response 1: Thanks your good suggestion. Authors have renamed the title" CaHsp18.1a, a small heat shock protein from pepper (Capsicum annuum L.), positively responds to heat, drought and salt tolerance ".
Ponit 2:I did not find any errors in the work but I do not understand why the Authors did not decide to carry out additional tests on the pepper cultivar which is commonly planted.
Response 2: Thank you for the excellent suggestion. To validate the function of certain gene (S), is to silence or overexpress in a plant system and to observe the phenotypic changes along with studying the biochemical indices. Since, expression and knockdown experiments were conducted in the pepper plant, the gene subcellular localization in the Tobacco and the overexpression in Arabidopsis. The overexpression in the system of pepper transgenic is not effective; the model plant Arabidopsis is a good choice to obtained transgenic lines in the third generations and to observe phenotypic changes along with other marker tests.
Ponit 3:The Authors did not clarify, using the abbreviation POD, whether they meant the sum of water soluble peroxidases (the ascorbate peroxidase is abbreviated properly).
Response 3: Here POD meant the sum of water soluble peroxidases.
Ponit 4:I would also like to draw the Authors' attention to very small markers in some graphs, e.g. in Figs. 3 and 5, in the legends. They are almost invisible when reading the paper version. Axis descriptions also have different sizes (please, compare the Fig. 3F and Fig. 3I).
Response 4: According to your kindly advice, we have modified the legends of Fig. 3 and 5, and resized the axis descriptions of Fig. 3F and Fig. 3I in the revised manuscript.
Ponit 5:The abreviation for a heat shock protein is written 'Hsp' and 'HSP' (e.g. in lines 75–77). This form should be unified.
Response 5: Thank you, we have unified as “HSP”, throughout the text in the revised manuscript.
Ponit 6:There are many minor editing errors, e.g. missing spaces or extra spaces in the manuscript. The entire text needs to be corrected.
Response 6: The entire text has been corrected in the revised manuscript.

Round 2
Reviewer 1 Report
I am sorry to inform you that in my opinion the authors did not adequately address my previous revision comments.
My main concerns are the same as I reported in my previous revision:
1) The manuscript is based on the physiological basis of abiotic stress response and tolerance in pepper plants. In principle this subject fits well with the journal (Horticulturae) aim.
However, a relevant part of the experimental results deal with experiments (overexpression or subcellular localization) performed on Arabidopsis or Tobacco which are not horticultural crops.
In their reply the authors state that “Since, the overexpression in the system of pepper transgenic is not effective, the model plant Arabidopsis is a good choice.”
I do not think that this is a good reason to include those experimental data in a manuscript submitted to “Horticulturae” and a different journal focusing on plant biochemistry, plant physiology or molecular biology may be a better choice.
2) Discussion should be improved as well as the english language.
3) Lastly, I will an additional comment.
In order to support their choice, in their reply the authors cite two previously published research articles by the same research group on a very similar subject.
The cited references are:
Feng, X.; Zhang, H.; Ali, M.; Gai, W.; Gong, Z., A small heat shock protein CaHsp25.9 positively regulates heat, salt, and drought stress tolerance in pepper (Capsicum annuum L.). Plant Physiology Biochemistry 2019, 142, 151-162.
Sun, J.; Cheng, G.; Huang, L.; Liu, S.; Ali, M.; Khan, A.; Yu, Q.; Yang, S.; Luo, D.; Gong, Z. Modified expression of a heat shock protein gene, CaHSP22.0, results in high sensitivity to heat and salt stress in pepper (Capsicum annuum L.). Scientia Horticulturae 2019, 249, 364-373.
Since all of these papers share exactly the same structure and very similar reported results, I would suggest that a manuscript to be published on “Horticulturae” had its own original structure.
This manuscript is a resubmission of an earlier submission. The following is a list of the peer review reports and author responses from that submission.
Round 1
Reviewer 1 Report
This paper delay with a small heat shock protein from pepper, Capsicum annuum.
They analyse the role of CaHsp18.1A in the response to heat, drought and salt.
They performed silencing by virus.
They did overexpression in Arabidopsis
They studied gene expression, MDA content, relative electrolyte leakage, enzymatic activity.
It is quite good.
I have thre major concerns
1/For GFP experiment, have they performed plasmolysis? Have they co labelled with FM424? We need that
2/we need REL in heat for silenced plants
3/ do they have expression levels of their gene in Arabidopsis lines
Abstract.
Line 16: you should introduce your protein of interest by one sentence. Why to focus on it? Explain
Line 19: what are the differences in responses between the lines.
Line 27: hat is “cell membrane” ? plasma membrane? Vesicle membrane?
Line 32. Delete “stresses from”
Line 52. Full name for ACD.
Line 93/ what do you mean by “was sensitive”. In fact delete and just mention the expression.
Line 94: define the lines. It was defined in abstract but you need to define them again here.
Line 95 add “promoter” before “contained”
Figure 1; in the text make it clear that the response is more intense and more early ine the R9 line.
Lines 224/225. Delete “which …. Roots”
Line 280: what is cell membrane? plasma membrane? Vesicle membrane? Have you perform plasmolysis? Have you co labelled with FM424?
Figure 3: we need REL in heat
Line 383. How do you explain such an heterogeneity between the overexpressing lines.
Many spaces are missing before reference brackets
Reviewer 2 Report
Revision to
Title: CaHsp18.1a, a small heat shock protein, positively responds to heat and drought and salt tolerance in pepper (Capsicum annuum L.)
Authors: YanLi Liu, Shuai Liu , JingJing Xiao , GuoXin Cheng , Saeed ul Haq , Zhen-Hui Gong
Abstract Journal: Horticulturae
Manuscript number: horticulturae-1148031
General remarks: The manuscript by Liu et al. describe the possible roles of CaHSP18.1, a member of the Heat Shock protein 20kDa from pepper, upon different stress treatments. The manuscript would be interesting, but is confusingly written. The idea would be promising but the results were not well evaluated and analyzed by the authors. Authors displayed a number of interesting experiments but did not focus their dissertation on any of these. A number of conclusions of the authors are speculations and were not directly demonstrated by results. In fact, these showed a number of incongruences.
Based on major and minor revision points (See below), the manuscript is rejected for publication on horticulturae.
Major points:
It is not clear why the expression of the CaHSP18.1 decrease at higher NaCl and mannitol concentrations? The author did not provide for any explanation. Moreover, the heat stress response of CaHSP18.1 is a results previous obtained by Guo et al., 2015.
Authors used two different pepper genotypes for this expression analysis. They argued “The present results showed that the accumulation of CaHsp20s can effectively reduce the damage caused by heat stress, prevent its irreversible polymerization by binding to denatured proteins, and enhance the heat resistance of pepper plants”. These speculations were not demonstrated with results of this manuscript.
To analyze silenced genotype, authors move soil-growing plants to hydroponic system but for me this is not appropriate system because this modification could induce different behaviors from the first growing system to the second. Anyway, the authors showed few parameters to validate their hypothesis.
The authors obtained a number of overexpressing genotypes treating them with high temperature, drought and salinity. The authors concluded, “Overexpression analyses of CaHsp18.1a in transgenic Arabidopsis further confirmed that CaHsp18.1a positively functions in responses to heat, drought, and salt stresses. The expression levels of other stress-related genes were also measured, and some were determined to be significantly affected by CaHsp18.1a over- expression”. This is absolutely not proved by the results of this manuscript. The ability of the genotypes to survive and/or tolerate adverse conditions is exclusively proved by photo. No numeric and statistical significant data were showed in the manuscript about this. Furthermore, drought and heat stressed plants were used of a number of analyses while salinity plants showed different and a reduced number of analyses.
Furthermore, the authors analyzed the expression of genes related to abiotic stress but in some case, not related to HSP. No explanations about the role of CaHsp18.1a in the regulation of these genes were discussed (only few references). In fact, overexpressed genotypes compared with wild type upon stress showed higher value of antioxidant enzymes expression and activities, higher expression of P5CS, higher expression of NCED, higher expression of MYB, higher expression of RD29. These are all symptoms of stress not the contrary
The authors argued “This indicated that CaHsp18.1a may improve stress resistance by enhancing the ability of pepper plants to remove ROS, but the specific mechanism needs further study”. This could be true but the authors should prove this by the evaluation of the oxidative stress and the ROS content. In this context, MDA content showed ambiguous results because both wild type and overexpressed genotypes showed a marked increase of lipid peroxidation. The slight differences compared wild type and overexpressed genotypes are not justified by the incredible enhanced expression and activities of scavenging enzymes.
Minor points:
Abstract: Thermophillic is absolutely a wrong term in this context.
Methods: Authors should specify the origin of the CaHSP18 sequence. The ID and/or the accession numbers of CaHSP18 from NCBI, Sol genomics network and other database should be added.
The experimental strategy description should be follow “Plant materials and growth conditions” section. Further, the authors should uniform the measurement units (mM/M).
Figure
Please organize Figure 1B-C-D replacing the roman numbers with the real sample times.